# The Kanerva Machine:
# A Generative Distributed Memory

**Yan Wu, Greg Wayne, Alex Graves, Timothy Lillicrap**
DeepMind
`{yanwu,gregwayne,gravesa,countzero}@google.com`

## Abstract

We present an end-to-end trained memory system that quickly adapts to new data and generates samples like them. Inspired by Kanerva's sparse distributed memory, it has a robust distributed reading and writing mechanism. The memory is analytically tractable, which enables optimal on-line compression via a Bayesian update-rule. We formulate it as a hierarchical conditional generative model, where memory provides a rich data-dependent prior distribution. Consequently, the top-down memory and bottom-up perception are combined to produce the code representing an observation. Empirically, we demonstrate that the adaptive memory significantly improves generative models trained on both the Omniglot and CIFAR datasets. Compared with the Differentiable Neural Computer (DNC) and its variants, our memory model has greater capacity and is significantly easier to train.

## 1 Introduction

Recent work in machine learning has examined a variety of novel ways to augment neural networks with fast memory stores. However, the basic problem of how to most efficiently use memory remains an open question. For instance, the slot-based external memory in models like Differentiable Neural Computers (DNCs Graves et al. (2016)) often collapses reading and writing into single slots, even though the neural network controller can in principle learn more distributed strategies. As as result, information is not shared across memory slots, and additional slots have to be recruited for new inputs, even if they are redundant with existing memories. Similarly, Matching Networks (Vinyals et al., 2016; Bartunov & Vetrov, 2016) and the Neural Episodic Controller (Pritzel et al., 2017) directly store embeddings of data. They therefore require the volume of memory to increase with the number of samples stored. In contrast, the Neural Statistician (Edwards & Storkey, 2016) summarises a dataset by averaging over their embeddings. The resulting "statistics" are conveniently small, but a large amount of information may be dropped by the averaging process, which is at odds with the desire to have large memories that can capture details of past experience.

Historically developed associative memory architectures provide insight into how to design efficient memory structures that store data in overlapping representations. For example, the Hopfield Net (Hopfield, 1982) pioneered the idea of storing patterns in low-energy states in a dynamic system. This type of model is robust, but its capacity is limited by the number of recurrent connections, which is in turn constrained by the dimensionality of the input patterns. The Boltzmann Machine (Ackley et al., 1985) lifts this constraint by introducing latent variables, but at the cost of requiring slow reading and writing mechanisms (i.e. via Gibbs sampling). This issue is resolved by Kanerva's sparse distributed memory model (Kanerva, 1988), which affords fast reads and writes and dissociates capacity from the dimensionality of input by introducing addressing into a distributed memory store whose size is independent of the dimension of the data[1].

In this paper, we present a conditional generative memory model inspired by Kanerva's sparse distributed memory. We generalise Kanerva's original model through learnable addresses and reparametrised latent variables (Rezende et al., 2014; Kingma & Welling, 2013; Bornschein et al., 2017). We solve the challenging problem of learning an effective memory writing operation by exploiting the analytic tractability of our memory model — we derive a Bayesian memory update rule

---

[1] For readers interested in the historical connection, we briefly review Kanerva's sparse distributed memory in Appendix B

that optimally trades-off preserving old content and storing new content. The resulting hierarchical generative model has a memory dependent prior that quickly adapts to new data, providing top-down knowledge in addition to bottom-up perception from the encoder to form the latent code representing data. As a generative model, our proposal provides a novel way of enriching the often over-simplified priors in VAE-like models (Rezende et al., 2016) through a adaptive memory. As a memory system, our proposal offers an effective way to learn online distributed writing which provides effective compression and storage of complex data.

## 2 BACKGROUND: VARIATIONAL AUTOENCODERS

Our memory architecture can be viewed as an extension of the variational autoencoder (VAE) (Rezende et al., 2014; Kingma & Welling, 2013), where the prior is derived from an adaptive memory store. A VAE has an observable variable $x$ and a latent variable $z$. Its generative model is specified by a prior distribution $p_\theta(z)$ and the conditional distribution $p_\theta(x|z)$. The intractable posterior $p_\theta(z|x)$ is approximated by a parameterised inference model $q_\phi(z|x)$. Throughout this paper, we use $\theta$ to represent the generative model's parameters, and $\phi$ to represent the inference model's parameters. All parameterised distributions are implemented as multivariate Gaussian distributions with diagonal covariance matrices, whose means and variances are outputs from neural networks as in (Rezende et al., 2014; Kingma & Welling, 2013).

We assume a dataset with independently and identically distributed (iid) samples $\mathcal{D} = \{x_1, \ldots, x_n, \ldots, x_N\}$. The objective of training a VAE is to maximise its log-likelihood $\mathbb{E}_{x \sim \mathcal{D}}[\ln p_\theta(x)]$. This can be achieved by jointly optimising $\theta$ and $\phi$ for a variational lower-bound of the likelihood (omitting the expectation over all $x$ for simplicity):

$$\mathcal{L} = \mathbb{E}_{q_\phi(z|x)}[\ln p_\theta(x|z)] - \mathrm{D_{KL}}(q_\phi(z|x) \| p_\theta(z)) \tag{1}$$

where the first term can be interpreted as the negative reconstruction loss for reconstructing $x$ using its approximated posterior sample from $q_\phi(z|x)$, and the second term as a regulariser that encourages the approximated posterior to be near the prior of $z$.

## 3 THE KANERVA MACHINE

To introduce our model, we use the concept of an *exchangeable* episode: $X = \{x_1, \ldots, x_t, \ldots, x_T\} \subset \mathcal{D}$ is a subset of the entire dataset whose order does not matter. The objective of training is the expected conditional log-likelihood (Bornschein et al., 2017),

$$\mathcal{J} = \int p(X, M) \ln p_\theta(X|M) \, \mathrm{d}M \mathrm{d}X = \int p(X) p(M|X) \sum_{t=1}^{T} \ln p_\theta(x_t|M) \, \mathrm{d}M \mathrm{d}X \tag{2}$$

The equality utilises the conditional independence of $x_t$ given the memory $M$, which is equivalent to the assumption of an exchangeable episode $X$ (Aldous, 1985). We factorise the joint distribution of $p(X, M)$ into the marginal distribution $p(X)$ and the posterior $p(M|X)$, so that computing $p(M|X)$ can be naturally interpreted as writing $X$ into the memory.

We propose this scenario as a general and principled way of formulating memory-based generative models, since $\mathcal{J}$ is directly related to the mutual information $I(X; M)$ through $I(X; M) = H(X) - H(X|M) = H(X) + \int p(X, M) \ln p_\theta(X|M) \, \mathrm{d}X \mathrm{d}M = H(X) + \mathcal{J}$. As the entropy of the data $H(X)$ is a constant, maximising $\mathcal{J}$ is equivalent to maximising $I(X; M)$, the mutual information between the memory and the episode to store.

### 3.1 THE GENERATIVE MODEL

We write the collection of latent variables corresponding to the observed episode $X$ as $Y = \{y_1, \ldots, y_t, \ldots, y_T\}$ and $Z = \{z_1, \ldots, z_t, \ldots, z_T\}$. As illustrated in Fig. 1 (left), the joint distribution of the generative model can be factorised as

$$p_\theta(X, Y, Z|M) = \prod_{t=1}^{T} p_\theta(x_t, y_t, z_t|M) = \prod_{t=1}^{T} p_\theta(x_t|z_t) p_\theta(z_t|y_t, M) p_\theta(y_t) \tag{3}$$

The first equality uses the conditional independence of $z_t, y_t, x_t$ given $M$, shown by the "plates" in Fig. 1 (left). The memory $M$ is a $K \times C$ random matrix with the matrix variate Gaussian distribution (Gupta & Nagar, 1999):

$$p(M) = \mathcal{MN}(R, U, V) \tag{4}$$

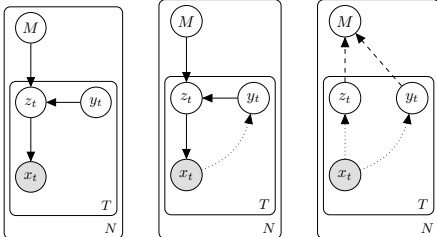

Figure 1: The probabilistic graphical model for the Kanerva Machine. Left: the generative model; Central: reading inference model. Right: writing inference model; Dotted lines show approximate inference and dashed lines represent exact inference.

where $R$ is a $K \times C$ matrix as the mean of $M$, $U$ is a $K \times K$ matrix that provides the covariance between rows of $M$, and $V$ is a $C \times C$ matrix providing covariances between columns of $M$. This distribution is equivalent to the multivariate Gaussian distribution of vectorised $M$: $p\left(\text{vec}\left(M\right)\right) = \mathcal{N}\left(\text{vec}\left(M\right) | \text{vec}\left(R\right), V \otimes U\right)$, where $\text{vec}\left(\cdot\right)$ is the vectorisation operator and $\otimes$ denotes the Kronecker product. We assume independence between the columns but not the rows of $M$, by fixing $V$ to be the identity matrix $I_C$ and allow the full degree of freedom for $U$. Since our experiments suggest the covariance between rows is useful for coordinating memory access, this setting balances simplicity and performance (Fig. 10).

Accompanying $M$ are the addresses $A$, a $K \times S$ real-value matrix that is randomly initialised and is optimised through back-propagation. To avoid degeneracy, rows of $A$ are normalised to have L2-norms of 1. The *addressing variable* $y_t$ is used to compute the weights controlling memory access. As in VAEs, the prior $p_\theta\left(y_t\right)$ is an isotropic Gaussian distribution $\mathcal{N}\left(\mathbf{0}, \mathbf{1}\right)$. A learned projection $b_t = f(y_t)$ then transforms $y_t$ into a $S \times 1$ key vector. The $K \times 1$ vector $w_t$, as weights across the rows of $M$, is computed via the product:

$$w_t = b_t{}^\intercal \cdot A = f(y_t)^\intercal \cdot A \tag{5}$$

The projection $f$ is implemented as a multi-layer perception (MLP), which transforms the distribution of $y_t$, as well as $w_t$, to potentially non-Gaussian distributions that may better suit addressing.

The code $z_t$ is a learned representation that generates samples of $x_t$ through the parametrised conditional distribution $p_\theta\left(x_t | z_t\right)$. This distribution is tied for all $t \in \{1 \ldots T\}$. Importantly, instead of the isotropic Gaussian prior, $z_t$ has a memory dependent prior:

$$p_\theta(z_t | y_t, M) = \mathcal{N}\left(z_t \,\middle|\, w_t{}^\intercal \cdot M, \sigma^2 \, I_C\right) \tag{6}$$

whose mean is a linear combination of memory rows, with the noise covariance matrix fixed as an identity matrix by setting $\sigma^2 = 1$. This prior results in a much richer marginal distribution, because of its dependence on memory and the addressing variable $y_t$ through $p_\theta(z_t | M) = \int p_\theta(z_t | y_t, M) p_\theta(y_t) \, \mathrm{d}y_t$.

In our hierarchical model, $M$ is a global latent variable for an episode that captures statistics of the entire episode (Bartunov & Vetrov, 2016; Edwards & Storkey, 2016), while the local latent variables $y_t$ and $z_t$ capture local statistics for data $x_t$ within an episode. To generate an episode of length $T$, we first sample $M$ once, then sample $y_t$, $z_t$, and $x_t$ sequentially for each of the $T$ samples.

### 3.2 THE READING INFERENCE MODEL

As illustrated in Fig. 1 (central), the approximated posterior distribution is factorised using the conditional independence:

$$q_\phi\left(Y, Z | X, M\right) = \prod_{t=1}^{T} q_\phi\left(y_t, z_t | x_t, M\right) = \prod_{t=1}^{T} q_\phi\left(z_t | x_t, y_t, M\right) q_\phi\left(y_t | x_t\right) \tag{7}$$

where $q_\phi\left(y_t | x_t\right)$ is a parameterised approximate posterior distribution. The posterior distribution $q_\phi\left(z_t | x_t, y_t, M\right)$ refines the (conditional) prior distribution $p_\theta(z_t | y_t, M)$ with additional evidence from $x_t$. This parameterised posterior takes the concatenation of $x_t$ and the mean of $p_\theta(z_t | y_t, M)$ (eq. 6) as input. The constant variance of $p_\theta(z_t | y_t, M)$ is omitted. Similar to the generative model, $q_\phi\left(y_t | x_t\right)$ is shared for all $t \in \{1 \ldots T\}$.

### 3.3 THE WRITING INFERENCE MODEL

A central difficulty in updating memory is the trade-off between preserving old information and writing new information. It is well known that this trade-off can be balanced optimally through Bayes' rule MacKay (2003). From the generative model perspective (eq. 2), it is natural to interpret memory writing as *inference* — computing the posterior distribution of memory $p(M|X)$. This section considers both batch inference — directly computing $p(M|X)$ and on-line inference — sequentially accumulating evidence from $x_1, \ldots, x_T$.

Following Fig. 1 (right), the approximated posterior distribution of memory can be written as

$$
\begin{aligned}
q_\phi\left(M|X\right) &= \int p_\theta\left(M, Y, Z|X\right) \, \mathrm{d}Z\mathrm{d}Y \\
&= \int p_\theta(M|\{y_1, \ldots, y_T\}, \{z_1, \ldots, z_T\}) \prod_{t=1}^{T} q_\phi(z_t|x_t) q_\phi(y_t|x_t) \, \mathrm{d}z_t\mathrm{d}y_t \quad (8) \\
&\approx p_\theta\left(M|\{y_1, \ldots, y_T\}, \{z_1, \ldots, z_T\}\right)\Big|_{y_t \sim q_\phi(y_t|x_t), z_t \sim q_\phi(z_t|x_t)}
\end{aligned}
$$

The last line uses one sample of $y_t$, $x_t$ to approximate the intractable integral. The posterior of the addressing variable $q_\phi(y_t|x_t)$ is the same as in section 3.2, and the posterior of code $q_\phi(z_t|x_t)$ is a parameterised distribution. We use the short-hand $p_\theta(M|Y, Z)$ for $p_\theta(M|\{y_1, \ldots, y_T\}, \{z_1, \ldots, z_T\})$ when $Y, Z$ are sampled as described here. We abuse notation in this section and use $Z = (z_1^\mathsf{T}; \ldots; z_T^\mathsf{T})$ as a $T \times C$ matrix with all the observations in an episode, and $W = (w_1^\mathsf{T}; \ldots; w_T^\mathsf{T})$ as a $T \times K$ matrix with all corresponding weights for addressing.

Given the linear Gaussian model (eq. 6), the posterior of memory $p_\theta(M|Y, Z)$ is analytically tractable, and its parameters $R$ and $U$ can be updated as follows:

$$
\Delta \leftarrow Z - W\,R \quad (9)
$$

$$
\Sigma_c \leftarrow W\,U \qquad\qquad \Sigma_z \leftarrow W\,U\,W^\mathsf{T} + \Sigma_\xi \quad (10)
$$

$$
R \leftarrow R + \Sigma_c^\mathsf{T}\,\Sigma_z^{-1}\,\Delta \qquad\qquad U \leftarrow U - \Sigma_c^\mathsf{T}\Sigma_z^{-1}\Sigma_c \quad (11)
$$

where $\Delta$ is the prediction error before updating the memory, $\Sigma_c$ is a $T \times K$ matrix providing the cross-covariance between $Z$ and $M$, $\Sigma_\xi$ is a $T \times T$ diagonal matrix whose diagonal elements are the noise variance $\sigma^2$ and $\Sigma_z$ is a $T \times T$ matrix that encodes the covariance for $z_1, \ldots, z_T$. This update rule is derived from applying Bayes' rule to the linear Gaussian model (Appendix E). The prior parameters of $p(M)$, $R_0$ and $U_0$ are trained through back-propagation. Therefore, the prior of $M$ can learn the general structure of the entire dataset, while the posterior is left to adapt to features presented in a subset of data observed within a given episode.

The main cost of the update rule comes from inverting $\Sigma_z$, which has a complexity of $\mathcal{O}(T^3)$. One may reduce the per-step cost via on-line updating, by performing the update rule using one sample at a time — when $X = x_t$, $\Sigma_z$ is a scalar which can be inverted trivially. According to Bayes' rule, updating using the entire episode at once is equivalent to performing the one-sample/on-line update iteratively for all observations in the episode. Similarly, one can perform intermediate updates using mini-batch with size between 1 and $T$.

Another major cost in the update rule is the storage and multiplication of the memory's row-covariance matrix $U$, with the complexity of $\mathcal{O}(K^2)$. Although restricting this covariance to diagonal can reduce this cost to $\mathcal{O}(K)$, our experiments suggested this covariance is useful for coordinating memory accessing (Fig. 10). Moreover, the cost of $\mathcal{O}(K^2)$ is usually small, since parameters of the model are dominated by the encoder and decoder. Nevertheless, a future direction is to investigating low-rank approximation of $U$ that better balance cost and performance.

### 3.4 TRAINING

To train this model, we optimise a variational lower-bound of the conditional likelihood $J$ (eq. 2), which can be derived in a fashion similar to standard VAEs:

$$
\begin{aligned}
\mathcal{L} = \mathbb{E}_{q_\phi(M|X)p(X)} \sum_{t=1}^{T} \Big\{ & \mathbb{E}_{q_\phi(y_t, z_t|x_t, M)}\left[\ln p_\theta\left(x_t|z_t\right)\right] \\
& -\mathrm{D}_{\mathrm{KL}}\left(q_\phi\left(y_t|x_t\right)\|p_\theta\left(y_t\right)\right) - \mathrm{D}_{\mathrm{KL}}\left(q_\phi\left(z_t|x_t, y_t, M\right)\|p_\theta\left(z_t|y_t, M\right)\right) \Big\}
\end{aligned} \quad (12)
$$

To maximise this lower bound, we sample $y_t, z_t$ from $q_\phi(y_t, z_t | x_t, M)$ to approximate the inner expectation. For computational efficiency, we use a mean-field approximation for the memory — using the mean $R$ in the place of memory samples (since directly sampling $M$ requires expensive Cholesky decomposition of the non-diagonal matrix $U$). Alternatively, we can further exploit the analytical tractability of the Gaussian distribution to obtain distribution-based reading and writing operations (Appendix F).

Inside the bracket, the first term is the usual VAE reconstruction error. The first KL-divergence penalises complex addresses, and the second term penalises deviation of the code $z_t$ from the memory-based prior. In this way, the memory learns useful representations that do not rely on complex addresses, and the bottom-up evidence only corrects top-down memory reading when necessary.

### 3.5 ITERATIVE SAMPLING

An important feature of Kanerva's sparse distributed memory is its iterative reading mechanism, by which output from the model is fed back as input for several iterations. Kanerva proved that the dynamics of iterative reading will decrease errors when the initial error is within a generous range, converging to a stored memory (Kanerva, 1988). A similar iterative process is also available in our model, by repeatedly feeding-back the reconstruction $\hat{x}_t$. This Gibbs-like sampling follows the loop in Fig. 1 (central). While we cannot prove convergence, in our experiments iterative reading reliably improves denoising and sampling.

To understand this process, notice that knowledge about memory is helpful in reading, which suggests using $q_\phi(y_t | x_t, M)$ instead of $q_\phi(y_t | x_t)$ for addressing (section 3.2). Unfortunately, training a parameterised model with the whole matrix $M$ as input can be prohibitively costly. Nevertheless, it is well-known in the coding literature that such intractable posteriors that usually arise in non-tree graphs (as in Fig. 1) can be approximated efficiently by loopy belief-propagation, as has been used in algorithms like Turbo coding (Frey & MacKay, 1998). Similarly, we believe iterative reading works in our model because $q_\phi(y_t | x_t)$ models the local coupling between $x_t$ and $y_t$ well enough, so iterative sampling with the rest of the model is likely to converge to the true posterior $q_\phi(y_t | x_t, M)$. Future research will seek to better understand this process.

## 4 EXPERIMENTS

Details of our model implementation are described in Appendix C. We use straightforward encoder and decoder models in order to focus on evaluating the improvements provided by an adaptive memory. In particular, we use the same model architecture for all experiments with both Omniglot and CIFAR dataset, changing only the the number of filters in the convolutional layers, memory size, and code size. We always use the on-line version of the update rule (section 3.3). The Adam optimiser was used for all training and required minimal tuning for our model (Kingma & Ba, 2014). In all experiments, we report the value of variational lower bound (eq. 12) $L$ divided by the length of episode $T$, so the per-sample value can be compared with the likelihood from existing models.

We first used the Omniglot dataset to test our model. This dataset contains images of hand-written characters with 1623 different classes and 20 examples in each class (Lake et al., 2015). This large variation creates challenges for models trying to capture the entire complex distribution. We use a $64 \times 100$ memory $M$, and a smaller $64 \times 50$ address matrix $A$. For simplicity, we always randomly sample 32 images from the entire training set to form an "episode", and ignore the class labels. This represents a worst case scenario since the images in an episode will tend to have relatively little redundant information for compression. We use a mini-batch size of 16, and optimise the variational lower-bound (eq. 12) using Adam with learning rate $1 \times 10^{-4}$.

We also tested our model with the CIFAR dataset, in which each $32 \times 32 \times 3$ real-valued colour image contains much more information than a binary omniglot pattern. Again, we discard all the label information and test our model in the unsupervised setting. To accommodate the increased complexity of CIFAR, we use convolutional coders with 32 features at each layer, use a code size of 200, and a $128 \times 200$ memory with $128 \times 50$ address matrix. All other settings are identical to experiments with Omniglot.

### 4.1 COMPARISON WITH VAEs

We first use the $28 \times 28$ binary Omniglot from Burda et al. (2015) and follow the same split of 24,345 training and 8,070 test examples. We first compare the training process of our model with a baseline

VAE model using the exact same encoder and decoder. Note that there is only a modest increase of parameters in the Kanerva Machine compared the VAE since the encoder and decoder dominates the model parameters.

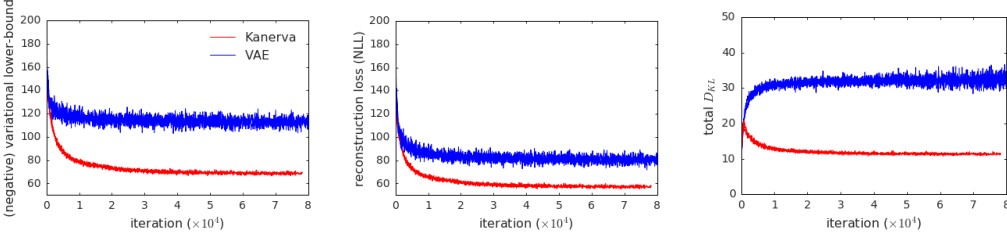

Figure 2: The negative variational lower bound (left), reconstruction loss (central), and KL-Divergence (right) during learning. The dip in the KL-divergence suggests that our model has learned to use the memory.

Fig. 2 shows learning curves for our model along with those for the VAE trained on the Omniglot dataset. We plot 4 randomly initialised instances for each model. The training is stable and insensitive to initialisation. Fig. 2 (left) shows that our model reached a significantly lower negative variational lower-bound versus the VAE. Fig. 2 (central) and (right) further shows that the Kanerva Machine achieved better reconstruction and KL-divergence. In particular, the KL-divergence of our model "dips" sharply from about the 2000th step, implying our model learned to use the memory to induce a more informative prior. Fig. 11 confirms this: the KL-divergence for $z_t$ has collapsed to near zero, showing that the top-down prior from memory $q_\phi(z_t|y_t, M)$ provides most of the information for the code. This rich prior is achieved at the cost of an additional KL-divergence for $y_t$ (Fig. 11, right) which is still much lower than the KL-divergence for $z_t$ in a VAE. Similar training curves are observed for CIFAR training (Fig. 12). Gemici et al. (2017) also observed such KL-divergence dips with a memory model. They report that the reduction in KL-divergence, rather than the reduction in reconstruction loss, was particularly important for improving sample quality, which we also observed in our experiments with Omniglot and CIFAR.

At the end of training, our VAE reached a negative log-likelihood (NLL) of $\leq 112.7$ (the lower-bound of likelihood), which is worse than the state-of-the-art unconditioned generation that is achieved by rolling out 80 steps of a DRAW model (NLL of 95.5, Rezende et al., 2016), but comparable to results with IWAE training (NLL of 103.4, Burda et al., 2015). In contrast, with the same encoder and decoders, the Kanerva Machine achieve conditional NLL of 68.3. It is not fair to directly compare our results with unconditional generative models since our model has the advantage of its memory contents. Nevertheless, the dramatic improvement of NLL demonstrates the power of incorporating an adaptive memory into generative models. Fig. 3 (left) shows examples of reconstruction at the end of training; as a signature of our model, the weights were well distributed over the memory, illustrating that patterns written into the memory were superimposed on others.

Figure 3: Left: reconstruction of inputs and the weights used in reconstruction, where each bin represents the weight over one memory slot. Weights are widely distributed across memory slots. Right: denoising through iterative reading. In each panel: the first column shows the original pattern, the second column (in boxes) shows the corrupted pattern, and the following columns show the reconstruction after 1, 2 and 3 iterations.

## 4.2 ONE-SHOT GENERATION

We generalise "one-shot" generation from a single image (Rezende et al., 2016), or a few sample images from a limited set of classes (Edwards & Storkey, 2016; Bartunov & Vetrov, 2016), to a

batch of images with many classes and samples. To better illustrate how samples are shaped by the conditioning data, in this section we use the same trained models, but test them using episodes with samples from only 2, 4 or 12 classes (omniglot characters)[2]. Fig. 4 compares samples from the VAE and the Kanerva Machine. While initial samples from our model (left most columns) are visually about as good as those from the VAE, the sample quality improved in consecutive iterations and the final samples clearly reflects the statistics of the conditioning patterns. Most samples did not change much after the 6th iteration, suggesting the iterative sampling had converged. Similar conditional samples from CIFAR are shown in Fig. 5. Notice that this approach, however, does not apply to VAEs, since VAEs do not have the structure we discussed in section 3.5. This is illustrated in Figure 8 by feeding back output from VAEs as input to the next iteration, which shows the sample quality did not improve after iterations.

Figure 4: One-shot generation given a batch of examples. The first panel shows reference samples from the matched VAE. Samples from our model conditioned on 12 random examples from the specified number of classes. Conditioning examples are shown above the samples. The 5 columns show samples after 0, 2, 4, 6, and 8 iterations.

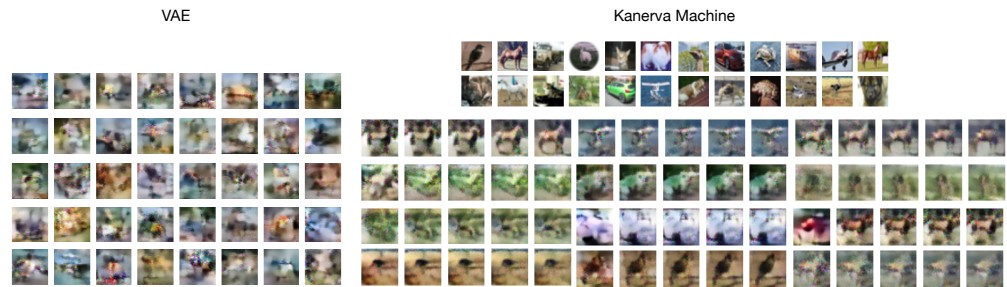

Figure 5: Comparison of samples from CIFAR. The 24 conditioning images (top-right) are randomly sampled from the entire CIFAR dataset, so they contains a mix of many classes. Samples from the matched VAE are blurred and lack meaningful local structure. On the other hand, samples from the Kanerva Machine have clear local structures, despite using the same encoder and decoder as the VAE. The 5 columns show samples after 0, 2, 4, 6, and 8 iterations.

## 4.3 DENOISING AND INTERPOLATION

To further examine generalisation, we input images corrupted by randomly positioned $12 \times 12$ blocks, and tested whether our model can recover the original image through iterative reading. Our model was not trained on this task, but Fig. 3 (right) shows that, over several iterations, input images can

---

[2]The Omniglot data from Burda et al. (2015) does not have label information, so for this experiment we produced our own labelled dataset by down-sampling the original Omniglot images (Lake et al., 2015) to $28 \times 28$ using the Python Image Library and then binarizing by thresholding at 20.

be recovered. Due to high ambiguity, some cases (e.g., the second and last) ended up producing incorrect but still reasonable patterns.

The structure of our model affords interpretability of internal representations in memory. Since representations of data $x$ are obtained from a linear combination of memory slots (eq. 6), we expect linear interpolations between address weights to be meaningful. We examined interpolations by computing 2 weight vectors from two random input images, and then linearly interpolating between these two vectors. These vectors were then used to read $z_t$ from memory (eq. 6), which is then decoded to produce the interpolated images. Fig. 7 in Appendix A shows that interpolating between these access weights indeed produces meaningful and smoothly changing images.

## 4.4 COMPARISON WITH DIFFERENTIABLE NEURAL COMPUTERS

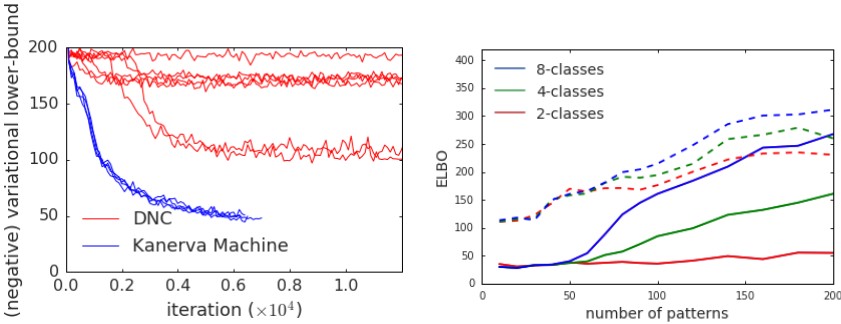

Figure 6: Left: the training curves of DNC and Kanerva machine both shows 6 instances with the best hyperparameter configuration for each model found via grid search. DNCs were more sensitive to random initilisation, slower, and plateaued with larger error. Right: the test variational lower-bounds of a DNC (dashed lines) and a Kanerva Machine as a function of different episode sizes and different sample classes.

This section compares our model with the Differentiable Neural Computer (DNC, Graves et al., 2016), and a variant of it, the Least Recently Used Architecture (LRUA, Santoro et al., 2016). We test these using the same episode storage and retrieval task as in previous experiments with Omniglot data. For a fair comparison, we fit the DNC models into the same framework, as detailed in Appendix D. Fig. 6 (left) illustrates the process of training the DNC and the Kanerva Machine. The LRUA did not passed the loss level of 150, so we did not include it in the figure. The DNC reached a test loss close to 100, but was very sensitive to hyper-parameters and random initialisation: only 2 out of 6 instances with the best hyper-parameter configuration (batch size = 16, learning rate= $3 \times 10^{-4}$) found by grid search reached this level. On the other hand, the Kanerva Machine was robust to these hyper-parameters, and worked well with batch sizes between 8 and 64, and learning rates between $3 \times 10^{-5}$ and $3 \times 10^{-4}$. The Kanerva Machine trained fastest with batch size 16 and learning rate $1 \times 10^{-4}$ and eventually converged below 70 test loss with all tested configurations. Therefore, the Kanerva Machine is significantly easier to train, thanks to principled reading and writing operations that do not depend on any model parameter.

We next analysed the capacity of our model versus the DNC by examining the lower bound of then likelihood when storing and then retrieving patterns from increasingly large episodes. As above, these models are still trained with episodes containing 32 samples, but are tested on much larger episodes. We tested our model with episodes containing different numbers of classes and thus varying amounts of redundancy. Fig. 6 (right) shows both models are able to exploit this redundancy, since episodes with fewer classes (but the same number of images) have lower reconstruction losses. Overall, the Kanerva Machine generalises well to larger episodes, and maintained a clear advantage over the DNC (as measured by the variational lower-bound).

## 5 DISCUSSION

In this paper, we present the Kanerva Machine, a novel memory model that combines slow-learning neural networks and a fast-adapting linear Gaussian model as memory. While our architecture is inspired by Kanerva's seminal model, we have removed the assumption of a uniform data distribution

by training a generative model that flexibly learns the observed data distribution. By implementing memory as a generative model, we can retrieve unseen patterns from the memory through sampling. This phenomenon is consistent with the observation of constructive memory neuroscience experiments (Hassabis et al., 2007).

Probabilistic interpretations of Kanerva's model have been developed in previous works: Anderson (1989) explored a conditional probability interpretation of Kanerva's sparse distributed memory, and generalised binary data to discrete data with more than two values. Abbott et al. (2013) provides an approximate Bayesian interpretation based on importance sampling. To our knowledge, our model is the first to generalise Kanerva's memory model to continuous, non-uniform data while maintaining an analytic form of Bayesian inference. Moreover, we demonstrate its potential in modern machine learning through integration with deep neural networks.

Other models have combined memory mechanisms with neural networks in a generative setting. For example, Li et al. (2016) used attention to retrieve information from a set of trainable parameters in a memory matrix. Notably, the memory in this model is not updated following learning. As a result, the memory does not quickly adapt to new data as in our model, and so is not suited to the kind of episode-based learning explored here. Bornschein et al. (2017) used discrete (categorical) random variables to address an external memory, and train the addressing mechanism, together with the rest of the generative model, though a variational objective. However, the memory in their model is populated by storing images in the form of raw pixels. Although this provides a mechanism for fast adaptation, the cost of storing raw pixels may be overwhelming for large data sets. Our model learns to to store information in a compressed form by taking advantage of statistical regularity in the images via the encoder at the perceptual level, the learned addresses, and Bayes' rule for memory updates.

Central to an effective memory model is the efficient updating of memory. While various approaches to learning such updating mechanisms have been examined recently (Graves et al., 2016; Edwards & Storkey, 2016; Santoro et al., 2016), we designed our model to employ an exact Bayes' update-rule without compromising the flexibility and expressive power of neural networks. The compelling performance of our model and its scalable architecture suggests combining classical statistical models and neural networks may be a promising direction for novel memory models in machine learning.

### ACKNOWLEDGMENTS

We would like to thank Sergey Bartunov, Charles Blundell, Jörg Bornschein, Karol Gregor, Shakir Mohamed, and Benigno Uria for helpful discussions, and to thank Dillon Graham and Jascha Sohl-Dickstein for pointing out mistakes in earlier manuscripts.

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

# APPENDIX

## A    EXTRA FIGURES

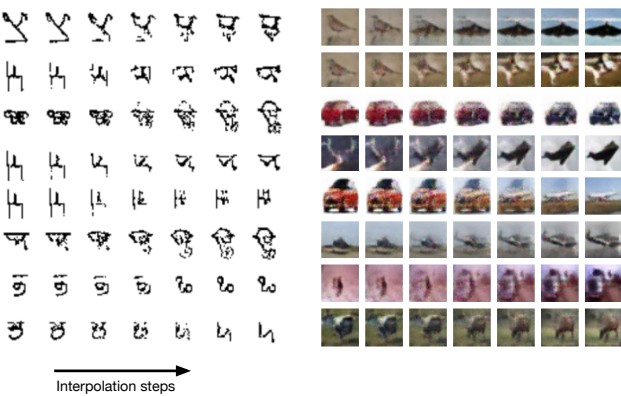

Figure 7:  Interpolation for Omniglot and CIFAR images. The first and last column show 2 random images from the data.  Between them are linear interpolations in the space of memory accessing weights $w_t$.

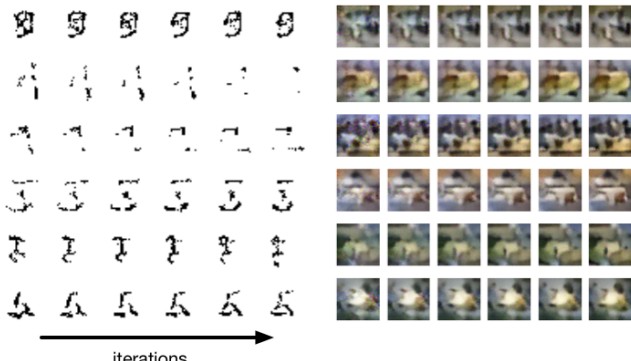

Figure 8: Iteratively sampled priors from VAE, for both Omniglot (left) and Cifar (right). In both panels, the columns show samples after 0, 2, 4, 6, 8 and 10 iterations, mirroring the procedure producing figure 4 and 5.

## B    SPARSE DISTRIBUTED MEMORY

This section reviews Kanerva's sparse distributed memory (Kanerva, 1988). For consistency with the rest of this paper, many of the notations are different from Kanerva's description. In contrast to many recent models, Kanerva's memory model is characterised by its distributed reading and writing operations. The model has two main components: a fixed table of addresses $A$ pointing to a modifiable memory $M$. Both $A$ and $M$ have the same size of $K \times D$, where $K$ is the number of addresses that and $D$ is the input dimensionality. Kanerva assumes all the inputs are uniform random vectors $y \in \{-1, 1\}^D$. Therefore, the fixed addresses $A_i$ are uniformly randomly sampled from $\{-1, 1\}^D$ to reflect the input statistics.

An input $y$ is compared with each address $A_k$ in $A$ through the Hamming distance. For binary vectors $a, b \in \{-1, 1\}^D$, the Hamming distance can be written as $h(a, b) = \frac{1}{2}(D - a \cdot b)$ where $\cdot$ represents inner product between two vectors. An address $k$ is *selected* when the hamming distance between $x$

and $A_k$ is smaller than a threshold $\tau$, so the selection can be summarised by the binary weight vector:

$$w_k = \begin{cases} 1, & h(x, A_k) \leqslant \tau \\ 0, & \text{otherwise} \end{cases} \tag{13}$$

During writing, a pattern $x$ is stored into $M$ by adding $M_k \leftarrow M_k + w_k\, x$. For reading, the memory contents pointed to by all the selected addresses are summed together to pass a threshold at 0 to produce a read out:

$$\hat{x} = \begin{cases} 1, & \sum_{k=1}^{K} w_k\, M_k > 0 \\ -1, & \text{otherwise} \end{cases} \tag{14}$$

This reading process can be iterated several times by repeatedly feeding-back the output $\hat{x}$ as input.

It has been shown analytically by Kanerva that when both $K$ and $D$ are large enough, a small portion of the addresses will always be selected, thus the operations are sparse and distributed. Although an address' content may be over-written many times, the stored vectors can be retrieved correctly. Moreover, Kanerva proved that even a significantly corrupted query can be discovered from the memory through iterative reading. However, the application of Kanerva's model is restricted by the assumption of a uniform and binary data distribution, on which Kanerva's analyses and bounds of performance rely (Kanerva, 1988). Unfortunately, this assumption is rarely true in practice, since real-world data typically lie on low-dimensional manifolds, and binary representation of data is less efficient in high-level neural network implementations that are heavily optimised for floating-point numbers.

## C  MODEL DETAILS

Figure 9 shows the architecture of our model compared with a standard VAE. For all experiments, we use a convolutional encoder to convert input images into $2C$ embedding vectors $e(x_t)$, where $C$ is the code size (dimension of $z_t$). The convolutional encoder has 3 consecutive blocks, where each block is a convolutional layer with $4 \times 4$ filter with stride 2, which reduces the input dimension, followed by a basic ResNet block without bottleneck (He et al., 2016). All the convolutional layers have the same number of filters, which is either 16 or 32 depending on the dataset. The output from the blocks is flattened and linearly projected to a $2C$ dimensional vector. The convolutional decoder mirrors this structure with transposed convolutional layers. All the "MLP" boxes in Fig. 9 are 2-layer multi-layer perceptron with ReLU non-linearity in between. We found that adding noise to the input into $q_\phi\,(y_t|x_t)$ helped stabilise training, possibly by restricting the information in the addresses. The exact magnitude of the added noise matters little, and we use Gaussian noise with zero mean and standard deviation of $0.2$ for all experiments. We use Bernoulli likelihood function for Omniglot dataset, and Gaussian likelihood function for CIFAR. To avoid Gaussian likelihood collapsing, we added uniform noise $\mathcal{U}(0, \frac{1}{256})$ to CIFAR images during training.

## D  DNC DETAILS

For a fair comparison, we wrap the differentiable neural computer (DNC) with the same interface as the Kanerva memory so that it can simply replace the memory $M$ in Fig. 9. More specifically, the DNC receives the addressing variable $y_t$ with the same size and sampled the same ways as described in the main text in reading and writing stages. During writing it also receives $z_t$ sampled from $q_\phi\,(z_t|x_t)$ as input, by concatenating $y_t$ and $z_t$ together as input into the memory controller.

Since DNCs do not have separated reading and writing stages, we separated this two process in our experiments: during writing, we discard the read-out from the DNC, and only keep its state as the memory; during reading, we discard the state at each step so it cannot be used for storing new information. In addition, we use a 2-layer MLP with 200 hidden neurons and ReLU nonlinearity as the controller instead of the commonly used LSTM to avoid the recurrent state being used as memory and interference with DNC's external memory. Another issue with off-the-shelf DNC (Graves et al., 2016; Santoro et al., 2016) is that controllers may generate output bypassing the memory, which can be particularly confusing in our auto-encoding setting by simply ignoring the memory and functioning as a skip connection. We avoid this situation by removing this controller output and ensure that the DNC only reads-out from its memory. Further, to focus on the memory performance, we remove

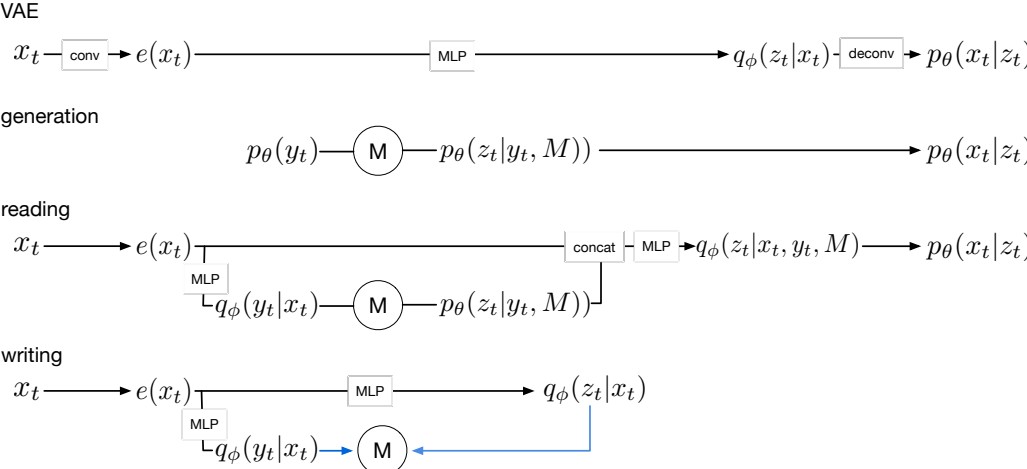

Figure 9: The architecture of the VAE and the Kanerva Machine used in our experiments. conv/deconv: convolutional and transposed convolutions neural networks. MLP: multiplayer perceptron. concat: vector concatenation. The blue arrows show memory writing as exact inference.

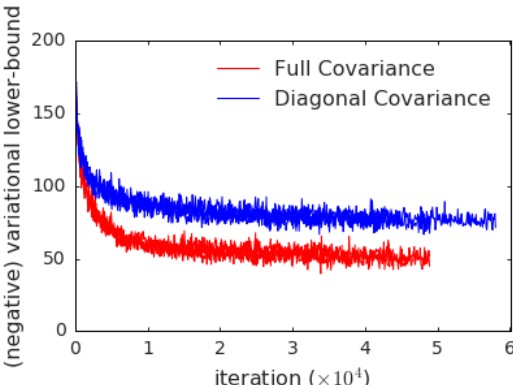

Figure 10: Covariance between memory rows is important. The two curves shows the test loss (negative variational lower bound) as a function of iterations. Four models using full $K \times K$ covariance matrix $U$ are shown by red curves and four models using diagonal covariance matrix are shown in blue. All other settings for these 8 models are the same (as described in section 4). These 8 models are trained on machines with similar setup. The models using full covariance matrices were slightly slower per-iteration, but the test loss decreased far more quickly.

the bottom-up stream in our model that compensates for the memory. This means directly sampling $z_t$ from $p_\theta(z_t|y_t, M)$, instead of $p_\theta(z_t|x_t, y_t, M)$, for the decoder $p_\theta(x_t|z_t)$, forcing the model to reconstruct solely using read-outs from the memory.

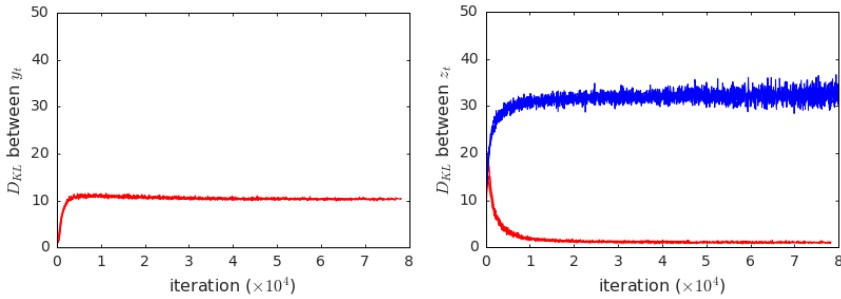

Figure 11: The KL-divergence between $y_t$ (left) and $z_t$ (right) during training.

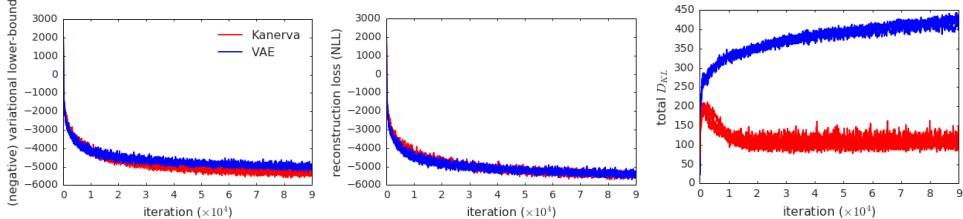

Figure 12: The negative variational lower bound, reconstruction loss, and total KL-divergence during CIFAR training. Although the difference between the lower bound objective is smaller than that during Omniglot training, the general patterns of these curves are similar to those in Fig. 2. The relatively small difference in KL-divergence significantly influences sample quality. Notice at the time of our submission, the training is continuing and the advantage of the Kanerva Machine over the VAE is increasing.

## E   DERIVATION OF THE ONLINE UPDATE RULE

Eq. 6 defines a linear Gaussian model. Using notations in the main paper, can write the joint distribution $p(\text{vec}(Z), \text{vec}(M)) = \mathcal{N}(\text{vec}(Z), \text{vec}(M); \mu_j, \Sigma_j)$, where

$$\mu_j = \begin{bmatrix} \text{vec}(WR) \\ \text{vec}(R) \end{bmatrix} \tag{15}$$

$$\Sigma_j = \begin{bmatrix} \Sigma_z \otimes I_C & \Sigma_c \otimes I_C \\ \Sigma_c^\mathsf{T} \otimes I_C & U \otimes I_c \end{bmatrix} \tag{16}$$

We can then use the conditional formula for the Gaussian to derive the posterior distribution $p(\text{vec}(M) | \text{vec}(Z)) = \mathcal{N}(\text{vec}(M); \mu_p, \Sigma_p)$, using the property Kronecker product:

$$\mu_p = \text{vec}(R) + \Sigma_c^\mathsf{T} \Sigma_z^{-1} \otimes I_C(\text{vec}(Z) - \text{vec}(WR)) \tag{17}$$

$$\Sigma_p = U \otimes I_c - \Sigma_c^\mathsf{T} \Sigma_z^{-1} \Sigma_c \otimes I_C \tag{18}$$

From properties of matrix variate Gaussian distribution, the above two equations can be re-arranged to the update rule in eq. 9 to 11.

## F   DISTRIBUTION-BASED READING AND WRITING

While the model we described in this paper works well using samples from $q_\phi(z_t|x_t)$ for writing to the memory (section 3.3) and the mean-field approximation during reading (section 3.4), here we describe an alternative that fully exploits the analytic tractability of the Gaussian distribution. To simplify notation, we use $\psi = \{R, U, V\}$ for all parameters of the memory.

For reading, eq. 6 can be replaced with a distribution that directly depends on $\psi$ through the integral:

$$p_\theta\left(z_t|y_t, \psi\right) = \int p_\theta\left(z_t|y_t, M\right) p(M) \, \mathrm{d}M$$
$$= \mathcal{N}\left(z_t| \, w_t^{\mathsf{T}} R, w \, U \, w^{\mathsf{T}} + \sigma^2 \, I_C\right) \tag{19}$$

For writing, the distribution $q_\phi\left(Z|X\right) = \prod_{t=1}^{T} q_\phi\left(z_t|x_t\right) = \mathcal{N}\left(\mu_Q, \Sigma_Q\right)$ ($\mu_Q$ and $\Sigma_Q$ are functions of $X$) can be incorporated into the Bayes' update rule by analytically marginalising-out $Z$:

$$p_\theta\left(M|Y, X\right) = \int p_\theta\left(M|Y, Z\right) q_\phi\left(Z|X\right) \, \mathrm{d}Z$$
$$= \int \frac{p_\theta\left(M\right) p_\theta\left(Z|Y, M\right)}{p_\theta\left(Z|Y\right)} q_\phi\left(Z|X\right) \, \mathrm{d}Z \tag{20}$$
$$\propto p_\theta\left(M\right) \int p_\theta\left(Z|Y, M\right) q_\phi\left(Z|X\right) \, \mathrm{d}Z$$

where we used Bayes' rule and dropped the normalising constant $p_\theta\left(Z|Y\right)$, and then replaced the equality with proportional-to accordingly. The last integral is:

$$\int p_\theta\left(Z|Y, M\right) q_\phi\left(Z|X\right) \, \mathrm{d}Z = \frac{1}{\sqrt{\det(2\pi\Sigma_{z'})}} \exp\left[-\frac{1}{2}(\mu_Q - WR)^{\mathsf{T}} \Sigma_{z'}^{-1}(\mu_Q - WR)\right]$$
$$\cdot \int p'(Z) \, \mathrm{d}Z \tag{21}$$
$$= \mathcal{N}\left(\mu_Q|WR, \Sigma_{z'}\right)$$

where $\Sigma_{z'} = W \, U \, W^{\mathsf{T}} + \Sigma_\xi + \Sigma_Q$ and $p'(Z)$ is a distribution of $Z$ whose exact form is unimportant. Therefore, eq. 20 shows that the posterior distribution of $M$ is proportional to the product between the prior $p_\theta\left(M\right)$ and the above likelihood term. From inspection, we can see the update rule (eq. 9 - 11) needs to be modified by replacing $\Sigma_z$ with $\Sigma_{z'}$ by adding the bottom-up uncertainty $\Sigma_Q$.

## G   DESCRIPTION OF THE ALGORITHM

---

**Algorithm 1** Iterative Reading

---
**Input:** Memory $M$, a (potentially noisy) query $x_t$, the number of iteration $n$
**Output:** An estimate of the noiseless $\hat{x}_t$
  initialise $i = 0$
  **while** $i < n$ **do**
    sample $y_t \sim q_\phi(y_t|x_t)$
    compute the key $b_t \leftarrow f(y_t)$
    Compute the weights $w_t \leftarrow b_t^{\mathsf{T}} \cdot A$
    read-out mean $\mu_z \leftarrow w_t^{\mathsf{T}} \cdot M$
    sample $z_t \sim q_\phi\left(z_t|x_t, y_t, M\right)$ which takes $\mu_z$ and $x_t$ as inputs
    sample the new query $x_t \sim p_\theta(x_t|z_t)$
    increment $i \leftarrow i + 1$
  **end while**
  return $\hat{x} \leftarrow x_t$

---

---

**Algorithm 2** Writing

---

**Input:** Images $\{x_t\}_{t=1}^T$, Memory $M$ with parameters $R$ and $U$
**Output:** Updated memory $M'$
  **for** each $y_t$ **do**
    sample $y_t \sim q_\phi(y_t|x_t)$
    compute the key $b_t \leftarrow f(y_t)$
    Compute the weights $w_t \leftarrow b_t^\mathsf{T} \cdot A$
    sample $z_t \sim q_\phi(z_t|x_t)$
    update parameters of $M$
      $\Delta \leftarrow Z - W\,R$
      $\Sigma_c \leftarrow W\,U$
      $\Sigma_z \leftarrow W\,U\,W^\mathsf{T} + \Sigma_\xi$
      $R \leftarrow R + \Sigma_c^\mathsf{T}\Sigma_z^{-1}\,\Delta$
      $U \leftarrow U - \Sigma_c^\mathsf{T}\Sigma_z^{-1}\Sigma_c$
  **end for**
  return $M'$ with the updated parameters $R$ and $U$

---

