# OpenReview forum: "The Kanerva Machine: A Generative Distributed Memory"
_ICLR.cc/2018/Conference — Accept (Poster)_

### Official Review · AnonReviewer2 · 2017-11-27
**Proposes a deep generative model where the prior distribution consists of a probabilistic memory. Neat idea, little discussion and comparison to related work.**

**Rating:** 6
**Confidence:** 4

**Review:**

The generative model comprises a real-valued matrix M (with a multivariate normal prior) that serves
as the memory for an episode (an unordered set of datapoints). For each datapoint a marginally independent
latent variable y_t is used to index into M and realize a conditional density
of another latent variable z. z_t is used to generate the data.

The proposal of learning with a probabilistic memory is interesting and the framework proposed is elegant and cleanly explained. The model is evaluated on the following tasks:
* Qualitative results on denoising and one-shot generation using the Omniglot dataset.
* Qualitative results on sampling from the model using the CIFAR dataset.
* Likelihood estimation on the Omniglot dataset

Questions and concerns:

The model appears novel and is interesting, the experiments, however, are lacking in that they
do not compare against other any recently proposed memory augmented deep generative models [Bornschein et al] and [Li et. al] (https://arxiv.org/pdf/1602.07416.pdf). At the very minimum, the paper should include a discussion and a comparison with the latter. Doing so will help better understand what is gained from using retaining a probabilistic form of memory versus a determinstic memory indexed with attention as in [Li et. al].

How does the model perform as a function of varying T (size of episodes) during training? It would be interesting to see how well the model performs in the limiting case of T=1.

What is the task being solved in Section 4.4 by the DNC and the Kanerva machine? Please state this in the main paper.

Training and Evaluation: There is a mismatch in the training and evaluation procedure the implications of which I don't
fully understand yet. The text states that the model was trained where each observation in an episode comprised randomly sampled datapoints. This corresponds to a generative process where (1) a memory is randomly drawn, (2) each observation in the episode is an independent draws from the memory conditioned decoder. During training,
points in an episode are randomly selected. At test time, (if I understand correctly, please correct me if I haven't), the model is evaluated by having multiple copies of the same test point within an episode. Is that correct? If so, doesn't that correspond to evaluating the model under a different generative assumption? Why is this OK?

Likelihood evaluation: Could you expand on how the ELBO of 68.3 is computed under the model for a single test image in the Omniglot dataset? The text says that the likelihood of each data-point was divided by T (the length of the episode considered). This seems at odds with models, such as DRAW, evaluate the likelihood -- once at the end of the generative drawing process. What is the per-pixel likelihood obtained on the CIFAR dataset and what is the likelihood on a model where T=1 (for omniglot/cifar)?

Using Labels: Following up on the previous point, what happens if labelled information from Omniglot or CIFAR is used to define points within an episode during the training procedure? Does this help or hurt performance?

For the denoising comparison, how do the results compare to those obtained if you simulate a Markov Chain (sample latent state conditioned on noisy image, sample latent state, sample denoised observation, repeat using denoised observation) using a VAE?

---

> ### Author Response · Authors · 2017-12-15
> **Response to comments**
>
> The model appears novel and is interesting, the experiments, however, are lacking in that they
> do not compare against other any recently proposed memory augmented deep generative models...the paper should include a discussion and a comparison with the latter...
>
> -- We agree that these works should be better highlighted in our manuscript.  We have added a new paragraph (3rd in the Discussion section) to describe the relations with these 2 papers.
> While both of these models share some commonalities with our model, they also have key differences which make direct experimental comparisons problematic.  As we describe in more detail in the Discussion, our paper addresses a different, although related, problem --- updating memory optimally.  As you mentioned, such update is not possible with the memory in Li et al., which is fixed over the course of episodes. Similarly, the likelihoods from our model have a very different meaning from Bornschein et al., since the only ambiguity in retrieving stored patterns in their model was in the categorical addressing variable; their model stores images in memory in the form of raw-pixels. We instead store compressed embeddings in a distributed fashion. As a result, the objective function we use (eq. 2) becomes a constant 0 for the model in Bornschein et al., since the mutual information between the memory and an episode of images I(X; M) is simply the entropy of these images H(X), when all these images are directly stored in the memory.
>
> How does the model perform as a function of varying T (size of episodes) during training? ... the limiting case of T=1.
>
> -- The performance under varying T is shown in figure 6 (right). There is a smooth rise of test loss with increasing T, and T=1 does not seem to be very different.
>
> What is the task being solved in Section 4.4 by the DNC and the Kanerva machine? Please state this in the main paper.
>
> -- We now clarify this in the main paper. It is the same episode storage and retrieval task as earlier in the paper, only we now look at the test regime where episode lengths are longer.
>
> Training and Evaluation: There is a mismatch in the training and evaluation procedure the implications of which I don't fully understand yet …
>
> -- With one exception there is no mismatch in training and evaluation, though we can now see where confusion may have crept in. We have revised the description in the paper to clarify this. In general, training and testing follow the same sampling procedure in constructing the episodes. The single exception is in the omniglot one-shot generation and comparison with DNC, where we control the number of Omniglot classes in an episode during testing for illustrative purpose only [i.e. to illustrate what happens with different levels of redundancy in the input data].  For all other comparisons the train and test losses and visualisation are identical.
>
> Likelihood evaluation: ... how the ELBO of 68.3 is computed … This seems at odds with models, such as DRAW … What is the per-pixel likelihood obtained on the CIFAR dataset...?
>
> -- “T” has different meanings in our model and in DRAW. DRAW is an autoregressive model that uses T steps to construct *one* image; in our model, T is the number of images, so we divide the total log-likelihood of T (conditionally) independent images by T for comparison. The likelihood of 5329 can be converted to the per-pixel bits 4.4973.
>
> --To get the number of 68.3, we first compute the ELBO for the conditional log-likelihood of an episode with 32 images, which is log P(x_1, x_2, … x_32 | M)  = 2185.6. Since log P(x_1, x_2, … x_32 | M) = log P(x_1|M) + log P(x_1|M) + ..+ log P(x_32|M) (conditional independence / exchangeability), we can compute the average ELBO for each image by dividing 2185.6 / 32 = 68.3.
>
> Using Labels: ... what happens if labelled information from Omniglot or CIFAR is used to define points within an episode during the training procedure?
>
> -- This is an interesting point. We think it will help performance, since the additional label information may help the model further reduce redundancy. We only trained on the worst case scenario without such information.
>
> How do the results compare to those obtained if you simulate a Markov Chain using a VAE?
>
> -- We tried iterative sampling using a VAE as well. However, iterative sampling did not improve performance with a VAE --- which is why, to the best of our knowledge, it has not been used in previous literature. In our model iterative sampling works because of the structure of the generative model (section 3.5). We now discuss this in the revision and illustrated it in a new figure (Figure 8 in the Appendix).
>
> Many thanks for the comments which have helped us improve the manuscript.  If you still feel that there are issues with the manuscript that would prevent you from raising your score, please point these out so that we can address them.

---

### Official Review · AnonReviewer3 · 2017-11-28
**Kanerva Machine Review**

**Rating:** 7
**Confidence:** 3

**Review:**

The paper presents the Kanerva Machine, extending an interesting older conceptual memory model to modern usage. The review of Kanerva’s sparse distributed memory in the appendix was appreciated. While the analyses and bounds of the original work were only proven when restricted to uniform and binary data, the extensions proposed bring it to modern domain of non-uniform and floating point data.

The iterative reading mechanism which provides denoising and reconstruction when within tolerable error bounds, whilst no longer analytically provable, is well shown experimentally.
The experiments and results on Omniglot and CIFAR provide an interesting insight to the model's behaviour with the comparisons to VAE and DNC also seem well constructed.

The discussions regarding efficiency and potential optimizations of writing inference model were also interesting and indeed the low rank approximation of U seems an interesting future direction.

Overall I found the paper well written and reintroduced + reframed a relatively underutilized but well theoretically founded model for modern use.

---

> ### Author Response · Authors · 2017-12-15
> **Response to comments**
>
> We appreciate your comments. Please let us know if you have any additional suggestions for the text or experiments that would further improve our paper, and potentially lead you to increase your score.

---

### Official Review · AnonReviewer1 · 2017-12-03
**Generalization of the distributed memory**

**Rating:** 7
**Confidence:** 2

**Review:**

This paper generalizes the sparse distributed memory model of Kanerva to the Kanerva Machine by formulating a variational generative model of episodes with memory as the prior.

Please discuss the difference from other papers that implement memory as a generative model, i.e. (Bornschein, Mnih, Zoran, Rezende 2017)

A probabilistic interpretation of Kanerva’s model was given before (Anderson, 1989 http://ieeexplore.ieee.org/document/118597/ ) and (Abbott, Hamrick, Griffiths, 2013). Please discuss.

I found the relation to Kanerva’s original model interesting and well explained. The original model was motivated by human long term memory and neuroscience. It would be nice if the authors can provide what neuroscience implications their work has, and comment on its biological plausibility.

---

> ### Author Response · Authors · 2017-12-15
> **Response to comments**
>
> Thank you for your comments.
>
> We agree these papers should be discussed and have added new text (paragraphs 2 and 3 in the Discussion) to describe this work.
>
> With regard to biological plausibility, we believe our main contribution is at the computational (rather than implementation) level: i.e. by providing a model that can be used in the context of complex memory tasks. As we focused on developing a functional and useful machine learning model, we don’t make any claims about biological plausibility beyond the relationship with Kanerva’s model, whose distributed structure was motivated by understanding of the brain.
>
> Please let us know if you have any additional suggestions for the text or experiments that would further improve our paper, and potentially lead you to increase your score.

---

### Decision · Program_Chairs · 2018-01-29
**ICLR 2018 Conference Acceptance Decision**

**Decision:**

Accept (Poster)

**Comment:**

This paper presents a distributed memory architecture based on a generative model with a VAE-like training criterion. The claim is that this approach is easier to train than other memory-based architectures. The model seems sound, and it is described clearly. The experimental validation seems a bit limited: most of the comparisons are against plain VAEs, which aren't a memory-based architecture. The discussion of "one-shot generalization" is confusing, since the task is modified without justification to have many categories and samples per category. The experiment of Section 4.4 seems promising, but this needs to be expanded to more tasks and baselines since it's the only experiment that really tests the Kanerva Machine as a memory architecture. Despite these concerns, I think the idea is promising and this paper contributes usefully to the discussion, so I recommend acceptance.